# Self-Supervised Weight Templates for Scalable Vision Model Initialization

Yucheng Xie [1 2]  Fu Feng [1 2]  Ruixiao Shi [1 2]  Jing Wang [1 2]  Yong Rui [1 2]  Xin Geng [1 2]

## Abstract

The increasing scale and complexity of modern model parameters underscore the importance of pre-trained models. However, deployment often demands architectures of varying sizes, exposing limitations of conventional pre-training and fine-tuning. To address this, we propose SWEET, a self-supervised framework that performs constraint-based pre-training to enable scalable initialization in vision tasks. Instead of pre-training a fixed-size model, we learn a shared weight template and size-specific weight scalers under Tucker-based factorization, which promotes modularity and supports flexible adaptation to architectures with varying depths and widths. Target models are subsequently initialized by composing and reweighting the template through lightweight weight scalers, whose parameters can be efficiently learned from minimal training data. To further enhance flexibility in width expansion, we introduce width-wise stochastic scaling, which regularizes the template along width-related dimensions and encourages robust, width-invariant representations for improved cross-width generalization. Extensive experiments on CLASSIFICATION, DETECTION, SEGMENTATION and GENERATION tasks demonstrate the state-of-the-art performance of SWEET for initializing variable-sized vision models.

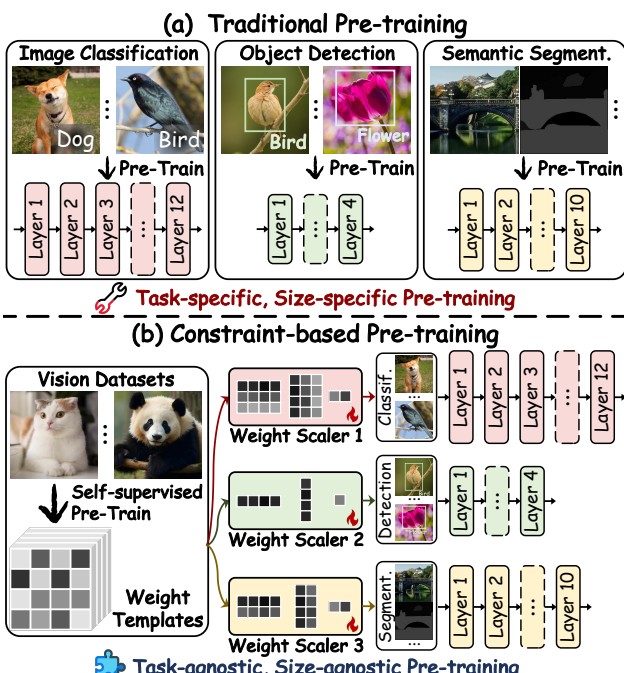

*Figure 1.* (a) Traditional pre-training paradigms produce fixed-size, task-specific models, which are difficult to adapt to downstream architectures with varying scales and task requirements. (b) SWEET adopts a constraint-based pre-training paradigm that extracts weight templates under structured constraints in a self-supervised manner, enabling flexible cross-scale and cross-task model initialization and efficient knowledge transfer.

## 1. Introduction

With the rapid growth of model scale, training from scratch has become increasingly inefficient (Liu et al., 2021; Wu et al., 2021), making pre-training a cornerstone of modern

[1]School of Computer Science and Engineering, Southeast University, Nanjing, China [2]Key Laboratory of New Generation Artificial Intelligence Technology and Its Interdisciplinary Applications (Southeast University), Ministry of Education, China. Correspondence to: Jing Wang <wangjing91@seu.edu.cn>, Xin Geng <xgeng@seu.edu.cn>.

*Proceedings of the $43^{rd}$ International Conference on Machine Learning*, Seoul, South Korea. PMLR 306, 2026. Copyright 2026 by the author(s).

visual learning, particularly in data-limited scenarios (Qiu et al., 2020; Han et al., 2021). However, traditional pre-training paradigms focus predominantly on maximizing performance on the pre-training dataset, producing models tightly coupled to a specific scale and downstream task domain (e.g., a ViT-L for image classification).

In practice, deployment is constrained by computational resources and downstream task requirements (Zhang et al., 2022), necessitating models of diverse scales and task-specific capabilities. Consequently, models that deviate from the configurations of off-the-shelf pre-trained models often necessitate retraining or knowledge distillation (Gou et al., 2021), imposing substantial computational overhead. Although several methods initialize downstream models of different scales by reusing or transforming pre-trained

weights for computational efficiency (Lan et al., 2020; Wang et al., 2022; 2023a; Xu et al., 2024), they often disrupt the structural coherence of pre-trained representations, leading to notable performance degradation.

Thus, recent advances shift pre-training from fixed-architecture optimization to decomposable parameter learning (Xie et al., 2025; 2026), enabling scalable downstream initialization. A representative method, WAVE (Feng et al., 2025b), formulates pre-training as a constraint-based optimization problem (Feng et al., 2026), training Vision Transformers under Kronecker-based constraints to learn structured **Weight Templates** instead of full model parameters. These templates are composed via the Kronecker product with lightweight **Weight Scalers**, enabling scalable model initialization with negligible computational overhead.

Despite its initial success, WAVE exhibits limitations that constrain flexible and extensible model initialization. Structurally, existing weight templates are restricted to homogeneous parameterizations, hindering parameter sharing across heterogeneous components such as attention heads and feed-forward modules. Furthermore, although they support width expansion, their fixed dimensionality limits adaptability to arbitrary widths. Task-wise, while effective for image classification, weight templates demonstrate limited transferability to heterogeneous tasks such as semantic segmentation and object detection. Together, these structural and task-related limitations prevent weight templates from serving as a universal initialization strategy for diverse visual models.

To address these limitations, we propose SWEET, a framework that learns self-supervised weight templates for scalable initialization across models of diverse sizes and visual tasks. Specifically, to facilitate parameter sharing across heterogeneous components, SWEET reorganizes and concatenates all parameters from different layers and modules into a unified weight matrix $\mathcal{W}$. Unlike prior constraint-based approaches (Feng et al., 2025b; Xie et al., 2024), SWEET reconstructs $\mathcal{W}$ via Tucker-based constraints (Malik et al., 2018) (see Eq. (5)), where $\mathcal{G}$ serves as the weight template and $(U, V, X)$ act as lightweight scalers, yielding a compact yet flexible parameter representation. A low-rank constraint is imposed on $\mathcal{G}$ as a "bottleneck" to condense size-agnostic knowledge (Feng et al., 2025a).

To capture task-agnostic knowledge across diverse vision tasks, SWEET employs *a self-supervised pre-training objective* to train weight templates, decoupling their representations from task-specific supervision and promoting generalizable visual features. Furthermore, to enhance adaptability to downstream models of arbitrary widths, we apply width-wise stochastic scaling during pre-training via dropout (Cai et al., 2020) on the weight scalers, improving the robustness of weight templates across diverse width configurations.

SWEET substantially reduces computational cost compared to conventional full-model pre-training, as its structured constraints and low-rank bottleneck filter non-transferable knowledge, enabling faster convergence while preserving transferable representations for scalable initialization and cross-task adaptation. Extensive experiments demonstrate SWEET's state-of-the-art performance in initializing variable-sized models across diverse vision tasks. On average across five model sizes, SWEET improves IMAGE CLASSIFICATION accuracy by 1.60%. Comparable advantages are observed on other vision tasks, with improvements of 2.04 AP, 2.76 mIoU, and 2.19 FID in OBJECT DETECTION, SEMANTIC SEGMENTATION, and IMAGE GENERATION, respectively.

Our contributions are as follows: 1) We propose SWEET, a self-supervised framework for pre-training structured weight templates that enable scalable initialization across varying model scales and diverse vision tasks. 2) We introduce width-wise stochastic scaling, a novel regularization strategy that enhances the robustness and adaptability of weight templates for initializing models with varying widths. 3) We establish a comprehensive benchmark for multi-scale model initialization across vision tasks, demonstrating that SWEET consistently outperforms existing methods in both cross-scale and cross-task model initialization.

**Conflict of Interest Disclosure.** The authors declare no financial or substantive conflicts of interest.

## 2. Related Work

### 2.1. Model Initialization

Model initialization is a fundamental factor influencing optimization efficiency and final performance (Narkhede et al., 2022; Hanin & Rolnick, 2018). Early methods relied on hand-crafted heuristics for random initialization (Glorot et al., 2010; Chen et al., 2021). With the rise of large-scale pre-training, initialization is now commonly inherited from pre-trained models, making fine-tuning the dominant paradigm (Qiu et al., 2020; Zhang et al., 2024).

To better leverage fixed-architecture pre-trained models for initializing models of varying sizes, several methods explore scalable initialization strategies. Mimetic Initialization (Trockman et al., 2023) leverages parameter patterns identified in pre-trained models to initialize new ones, while GHN (Knyazev et al., 2021; 2023) predict target model parameters using a graph hypernetwork conditioned on the computational graph. Weight Selection (Xu et al., 2024) directly transfers selected parameters from larger models to smaller ones. Despite these advances, representations in conventionally pre-trained models are entangled within parameter matrices, making direct splitting or transforma-

tion prone to negative transfer due to parameter mismatches or feature disruption. Our SWEET addresses this by pre-training structured weight templates rather than full models, enabling scalable initialization across model sizes.

## 2.2. Learngene and Weight Templates

LEARNGENE (Feng et al., 2025a; Wang et al., 2023b) is a biologically inspired knowledge transfer paradigm that encapsulates size-agnostic knowledge into modular neural units, termed learngenes, facilitating efficient adaptation across model scales. Early learngene methods primarily operate on a layer-wise basis. Heur-LG (Wang et al., 2022) identifies learngenes as layers with minimal gradient variation during continual learning, while Auto-LG (Wang et al., 2023b) selects layers whose representations best align with the target network via meta-learning. TLEG (Xia et al., 2024) models learngenes as pairs of base layers that can be linearly combined to initialize models of varying depths.

WAVE (Feng et al., 2025b) advances this line of work by introducing Weight Templates, overcoming layer-wise constraints by representing each weight matrix as a weighted combination of concatenated templates. Our SWEET extends this paradigm with *self-supervised weight templates*, substantially enhancing initialization flexibility and enabling universal model initialization across visual tasks.

# 3. Methods

## 3.1. Preliminaries

### 3.1.1. MASKED AUTOENCODERS (MAE)

MAE (He et al., 2022) is a self-supervised pre-training framework based on ViTs that learns visual representations by reconstructing randomly masked image patches. Given an input image, a large portion of patches (typically 75%) is masked, and the encoder processes only the visible patches to produce latent representations. A lightweight decoder then reconstructs the masked patches, optimized via

$$\mathcal{L}_{\text{MAE}} = \frac{1}{|\mathcal{M}|} \sum_{i \in \mathcal{M}} \|\hat{x}_i - x_i\|_2^2 \qquad (1)$$

where $\mathcal{M}$ is the set of masked patches, and $x_i$ and $\hat{x}_i$ denote the original and reconstructed patch embeddings. This self-supervised objective encourages the encoder to extract fundamental, generalizable visual features that are broadly transferable across downstream vision tasks.

### 3.1.2. VISION TRANSFORMER (VIT)

ViT (Dosovitskiy et al., 2021) comprises $L$ stacked layers, each containing a multi-head self-attention (MSA) followed by a multi-layer perceptron (MLP). In MSA, $h$ attention heads process the input, and their concatenated outputs are

projected using a learnable weight matrix $W_o \in \mathbb{R}^{hd \times D}$:

$$\text{MSA} = \text{concat}(A_1, A_2, ..., A_h)W_o, \ W_o \in \mathbb{R}^{hd \times D} \quad (2)$$

Within a single attention head $A_i$, queries $Q_i$, keys $K_i$, and values $V_i \in \mathbb{R}^{N \times d}$ are obtained via learnable projections $W_q^i$, $W_k^i$, and $W_v^i \in \mathbb{R}^{D \times d}$, and self-attention is given by

$$A_i = \text{softmax}\Big(\frac{Q_i K_i^\top}{\sqrt{d}}\Big)V_i, \quad A_i \in \mathbb{R}^{N \times d} \qquad (3)$$

where $N$ is the number of input patches, $D$ is the patch embedding dimension, and $d$ is the attention head dimension, typically $D = hd$ in standard multi-head self-attention.

MLP consists of two linear projections $W_{\text{in}} \in \mathbb{R}^{D \times D'}$ and $W_{\text{out}} \in \mathbb{R}^{D' \times D}$ with a GELU (Hendrycks et al., 2016) activation, formulated as:

$$\text{MLP}(x) = \text{GELU}(xW_{\text{in}} + b_1)W_{\text{out}} + b_2 \qquad (4)$$

where $b_1$, $b_2$ are bias and $D'$ is the hidden layer dimension, which is typically set to $D' = 4D$ in standard ViT.

## 3.2. Tucker-based Weight Template

SWEET learns structured weight templates through a constraint-based pre-training paradigm, in which structured constraints are imposed to learn templates instead of a full model, enabling scalable and regularized initialization. Representative works such as WAVE (Feng et al., 2025b) assign dedicated templates to individual components, focusing primarily on cross-scale knowledge sharing while overlooking shared patterns across components (e.g., between MSA and FFN), thereby limiting the generality of templates.

To address this limitation, we first aggregate the primary weight matrices of an $L$-layer ViT, $\theta = \{W_q^{(1 \sim L)}, W_k^{(1 \sim L)}, W_v^{(1 \sim L)}, W_o^{(1 \sim L)}, W_{\text{in}}^{(1 \sim L)}, W_{\text{out}}^{(1 \sim L)}\}^1$, into a unified weight matrix $\mathcal{W} \in \mathbb{R}^{L \times P}$, where each row represents a layer and $P = D \cdot (4hd + 2D') = 12D \cdot D$, thereby bridging the boundaries between heterogeneous model components.

Next, we impose structured constraints on $\mathcal{W}$. Instead of Kronecker (Feng et al., 2025b; 2026) or SVD-based constraints (Xie et al., 2024), we adopt a more general Tucker-based constraint. Specifically, the unified weight matrix $\mathcal{W}$ is reconstructed as

$$\mathcal{W} \Leftarrow \mathcal{G} \times_1 X \times_2 U \times_3 V, \qquad (5)$$

Here, $\mathcal{G} \in \mathbb{R}^{r_1 \times r_2 \times r_3}$ is the core tensor, serving as a universal *Weight Template* encoding size-agnostic knowledge, while the lightweight *Weight Scalers* $(X, U, V)$ modulate its reuse and composition to reconstruct $\mathcal{W}$, with

---

[1] $W_q^{(1 \sim L)}$ denotes $W_q^{(1)}, \ldots, W_q^{(L)}$ for brevity

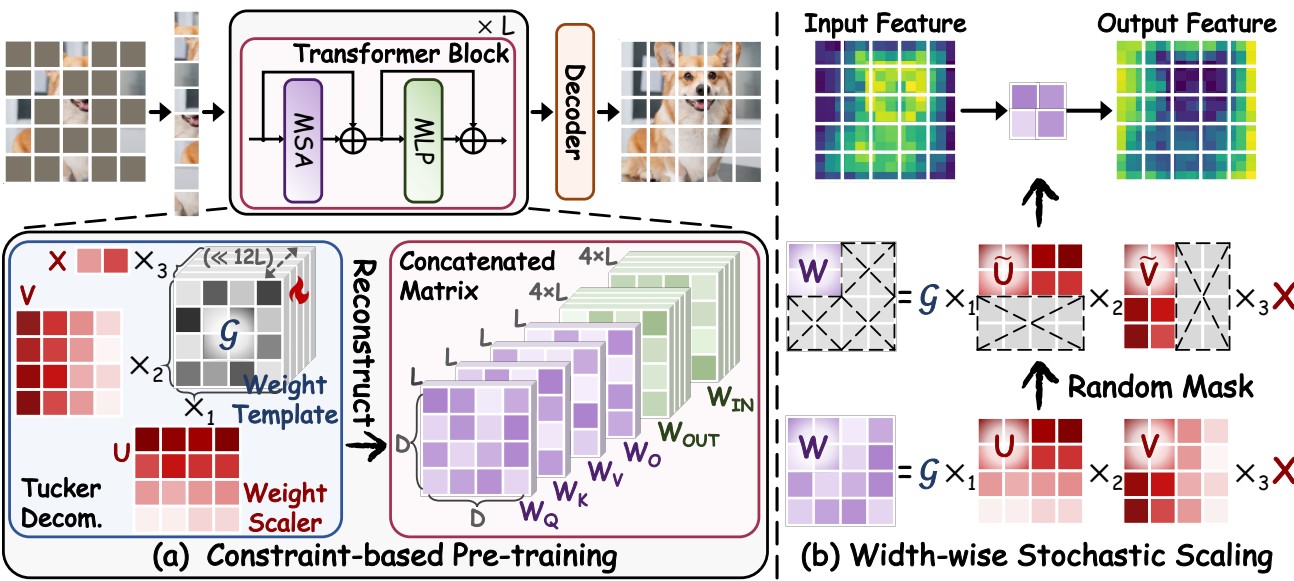

*Figure 2.* **Overview of SWEET. (a) Constraint-based Pre-training** of the weight template $\mathcal{G}$ with weight scalers $(U, V, X)$, where Tucker- and low-rank constraints are applied to condense size-agnostic knowledge within the template. A self-supervised objective guides pre-training to learn generalizable visual representations, enhancing the template's universality across diverse vision tasks. **(b) Width-wise Stochastic Scaling** randomly masks weight scalers along the width during pre-training, discouraging overfitting to a specific width and promoting the organization of template knowledge for flexible adaptation across models of varying widths.

$X \in \mathbb{R}^{12L \times r_1}$, $U \in \mathbb{R}^{D \times r_2}$, $V \in \mathbb{R}^{D \times r_3}$. The operator $\times_i$ denotes mode-$i$ tensor-matrix multiplication.

To capture size-agnostic knowledge, we impose a low-rank constraint on the weight template $\mathcal{G}$, with $r_1 \times r_2 \times r_3 \ll L \times P$, following (Feng et al., 2025a;b; Xie et al., 2024). Acting as a bottleneck (Zador, 2019), this constraint concentrates reusable information within $\mathcal{G}$, filtering knowledge that poorly transfers across model sizes and thereby enhancing template transferability and pre-training efficiency.

### 3.3. Self-Supervised Pre-Training and Width-Wise Stochastic Scaling

Early weight template training methods (Feng et al., 2025b; Xie et al., 2024) optimize conventional cross-entropy or reconstruction losses on image classification or generation, restricting their applicability to specific vision tasks and preventing their use as universal visual model initializers.

To capture generalizable visual knowledge beyond task-specific patterns, SWEET trains weight templates in a self-supervised manner. Specifically, after aggregating the weights of an $L$-layer ViT as in Eq. (5), SWEET jointly optimizes the weight template $\mathcal{G}$ and scalers $(X, U, V)$, through which the full model parameters are implicitly reconstructed. Formally, the pre-training objective is

$$\min_{\mathcal{G}, U, V, X} \mathcal{L}_{\text{MAE}}\big(f_\theta(x)\big),$$
$$\text{s.t. } \text{concat}(\theta) = \mathcal{W} = \mathcal{G} \times_1 X \times_2 U \times_3 V. \quad (6)$$

where $\mathcal{L}_{\text{MAE}}$ is the reconstruction loss defined in Eq. (1). This indirect optimization decouples knowledge extraction from specific parameter values, regularizes the initialization space, and promotes the learning of universal and structural visual patterns (see Algorithm 1).

Traditional layer-based learngene methods (Wang et al., 2022; 2023b; Xia et al., 2024) struggle to generalize across model widths, as their transfer units are intrinsically tied to fixed layers, while WAVE is constrained by the fixed dimensionality of its weight templates. To support flexible width expansion, SWEET introduces **width-wise stochastic scaling**, which applies structured dropout to the weight scalers, encouraging the weight template to capture width-robust knowledge:

$$\tilde{U} = M_U \odot U, \quad \tilde{V} = M_V \odot V \quad (7)$$

where $M_U$ and $M_V$ are independently sampled binary masks, drawn from a predefined distribution over width configurations, and $\odot$ denotes element-wise multiplication. The reconstructed weight matrix is

$$\mathcal{W} \Leftarrow \mathcal{G} \times_1 X \times_2 \tilde{U} \times_3 \tilde{V}. \quad (8)$$

By introducing stochastic scaling during pre-training, the model is prevented from overfitting to a fixed width, forcing it to reorganize knowledge along the width dimension. This reinforces width-invariant structural representations in the low-index dimensions of the weight template, enabling stable adaptation to models with varying widths.

Following (Li & He, 2025), to enhance generalization, we incorporate several architectural enhancements, including SwiGLU (Shazeer, 2020), RMSNorm (Zhang & Sennrich, 2019), and RoPE, originally proposed for language models.

### 3.4. Scalable Model Initialization

Benefiting from structured constraints and a low-rank bottleneck that filters non-transferable knowledge, constraint-based pre-training is substantially efficient than conventional full-model pre-training and incurs a once-for-all cost. The learned weight template $\mathcal{G}$ enables zero- or negligible-cost initialization of models with arbitrary sizes. During initialization, the template is kept frozen, while the lightweight scalers $(X, U, V)$ are directly selected or randomly initialized to match the target scale and optionally optimized, enabling flexible and scalable model instantiation.

Specifically, given a downstream model with parameters $\theta_\star$, we form a unified weight matrix $\mathcal{W}_\star = \text{concat}(\theta_\star) \in \mathbb{R}^{L_\star \times P_\star}$ by concatenating its layer-wise parameters, where $L_\star$ is the number of layers and $P_\star$ denotes the total per-layer width of the concatenated weight matrices. The weight scalers are ***randomly initialized*** or ***directly inherited*** from pre-trained $(X, U, V)$ to fit the target dimensions, producing $X_\star \in \mathbb{R}^{12L_\star \times r_1}$, $U_\star \in \mathbb{R}^{D_\star \times r_2}$, and $V_\star \in \mathbb{R}^{D_\star \times r_3}$ as above, which reconstruct $\mathcal{W}_\star$ while retaining the size-agnostic knowledge embedded in the frozen template $\mathcal{G}$.

Benefiting from width-wise stochastic scaling, target models with modest scale variations can be effectively initialized by directly selecting dimension-aligned slices from pre-trained $(X, U, V)$. For extremely compact models, template knowledge can be further adapted by lightly optimizing the scalers on a small dataset, while keeping the template $\mathcal{G}$ frozen:

$$\min_{U_\star, V_\star, X_\star} \quad \mathcal{L}_{\text{MAE}}\big(f_{\theta_\star}(x)\big),$$
$$\text{s.t.} \quad \text{concat}(\theta_\star) = \mathcal{W}_\star = \mathcal{G} \times_1 X_\star \times_2 U_\star \times_3 V_\star. \tag{9}$$

Owing to the small size of the weight scalers—typically only a few thousand parameters—optimization converges within a few hundred iterations ($\approx 0.16$ epoch), imposing negligible computational overhead. Once learned, the target model is initialized via Eq. (5), after which it can be trained conventionally without additional constraints.

## 4. Experiments

### 4.1. Experimental Setup

**Vision Tasks and Datasets**  SWEET is first pre-trained in a self-supervised manner on ImageNet-1K (Deng et al., 2009) to learn the weight template, and is subsequently evaluated on four representative vision tasks: IMAGE CLASSIFICATION and IMAGE GENERATION on ImageNet-1K, SEMANTIC SEGMENTATION on ADE20K (Zhou et al., 2019),

and OBJECT DETECTION on COCO (Lin et al., 2014). Additional details are provided in Appendix A.2.

**Network Structures**  We adopt ViT-Base (ViT-B/16) (Dosovitskiy et al., 2021) as the backbone for weight template pre-training. To evaluate SWEET's scalable initialization across depth and width, we consider ViT configurations with depths $L \in \{3, 6, 12\}$ and widths adjusted via the number of attention heads, $H \in \{3, 6, 12\}$.

**Evaluation Metrics**  We adopt standard metrics for each vision task. IMAGE CLASSIFICATION is evaluated using Top-1 and Top-5 accuracy. IMAGE GENERATION is assessed via Fréchet Inception Distance (FID) (Heusel et al., 2017) and Inception Score (IS) (Salimans et al., 2016) to capture visual fidelity and diversity. OBJECT DETECTION is measured by mean Average Precision for bounding boxes and masks ($AP^{\text{box}}$, $AP^{\text{mask}}$), while SEMANTIC SEGMENTATION is quantified by mean Intersection over Union (mIoU) and mean pixel accuracy (mAcc).

**Training Details**  The weight template is pre-trained in a self-supervised manner for 450 epochs on a batch size of 1024, using AdamW with a learning rate of $6 \times 10^{-4}$ and a cosine learning rate scheduler on an NVIDIA RTX 4090 GPU. In comparison, baseline methods use the official MAE weights from (He et al., 2022), traditionally pre-trained for 800 epochs with a batch size of 4096, making SWEET significantly more computationally efficient. Additional details are provided in Appendix A.3.

### 4.2. Baselines

We compare SWEET with state-of-the-art scalable model initialization methods. 1) WT-Select (Xu et al., 2024) initializes target models by directly selecting and reusing weight subsets from a pre-trained model according to predefined rules. 2) DMAE (Bai et al., 2023) distills knowledge from a teacher model by training student models to reconstruct masked inputs and align their intermediate feature maps. 3) Isomorphic Pruning (Fang et al., 2024) partitions parameters based on computational topology and ranks them within groups to guide pruning. 4) WAVE (Feng et al., 2025b), a representative constraint-based method, constructs weight templates via Kronecker-based constraints.

## 5. Results

### 5.1. Performance on Discriminative Vision Tasks

#### 5.1.1. IMAGE CLASSIFICATION

We first evaluate SWEET on IMAGE CLASSIFICATION, a standard vision task whose accuracy is highly sensitive to low- and mid-level features, making it a reliable measure

*Table 1.* Scalable initialization performance of SWEET across model scales on fundamental vision tasks. $L_l H_h$ denotes models with $l$ layers and $h$ attention heads, corresponding to the model's depth and width, respectively. "Para.(M)" and "FLOPs (G)" indicate the parameter count and computational complexity for each model scale. Models for IMAGE CLASSIFICATION and OBJECT DETECTION are trained for 30 epochs, while SEMANTIC SEGMENTATION are trained for 160K iterations after initialization.

| | $L_3H_{12}$ | | $L_6H_{12}$ | | $L_6H_6$ | | $L_{12}H_6$ | | $L_{12}H_3$ | | Average | |
|---|---|---|---|---|---|---|---|---|---|---|---|---|
| Para./FLOPs | 22.8M / 8.6G | | 44.0M / 17.0G | | 11.4M / 4.3G | | 22.1M / 8.5G | | 5.7M / 2.2G | | | |
| **IMAGE CLASSIFICATION** | Top1 | Top5 | Top1 | Top5 | Top1 | Top5 | Top1 | Top5 | Top1 | Top5 | Top1 | Top5 |
| WT-Select (Xu et al., 2024) | 66.44 | 86.46 | 75.91 | 92.42 | 67.03 | 86.93 | 66.51 | 86.81 | 43.49 | 68.58 | *63.87* | *84.24* |
| DMAE (Bai et al., 2023) | 66.67 | 86.71 | 73.95 | 91.17 | 64.86 | 85.65 | 65.70 | 86.17 | 49.07 | 73.77 | *64.05* | *84.69* |
| Iso. Pruning (Fang et al., 2024) | 66.46 | 86.35 | 75.85 | 92.54 | 68.25 | 88.00 | 70.86 | 89.67 | 44.53 | 68.89 | *65.19* | *85.09* |
| WAVE (Feng et al., 2025b) | 63.75 | 84.91 | 76.97 | 93.19 | 68.38 | 88.16 | 71.27 | 90.01 | 55.56 | 79.35 | *67.19* | *87.12* |
| SWEET | **67.21** | **86.93** | **77.42** | **93.25** | **70.34** | **89.22** | **71.70** | **90.24** | **57.28** | **80.31** | **68.79** | **87.99** |
| | ↑0.54 | ↑0.22 | ↑0.45 | ↑0.06 | ↑1.96 | ↑1.06 | ↑0.42 | ↑0.23 | ↑1.72 | ↑0.96 | *↑1.60* | *↑0.87* |
| **OBJECT DETECTION** | AP^box | AP^mask | AP^box | AP^mask | AP^box | AP^mask | AP^box | AP^mask | AP^box | AP^mask | AP^box | AP^mask |
| WT-Select (Xu et al., 2024) | 27.02 | 25.89 | 36.53 | 33.91 | 25.02 | 23.89 | 34.09 | 31.40 | 23.72 | 22.66 | *29.28* | *27.55* |
| DMAE (Bai et al., 2023) | 25.55 | 24.29 | 29.85 | 27.80 | 25.16 | 23.88 | 32.82 | 30.20 | 26.51 | 24.95 | *27.98* | *26.22* |
| Iso. Pruning (Fang et al., 2024) | 26.97 | 25.75 | 36.64 | 33.94 | 23.59 | 22.54 | 35.74 | 32.85 | 23.13 | 21.96 | *29.21* | *27.41* |
| WAVE (Feng et al., 2025b) | 27.94 | 26.63 | 34.19 | 31.91 | 26.14 | 24.93 | 33.55 | 31.15 | 25.94 | 24.45 | *29.55* | *27.81* |
| SWEET | **28.88** | **27.23** | **38.16** | **35.16** | **27.33** | **25.80** | **36.13** | **33.02** | **27.46** | **25.82** | **31.59** | **29.41** |
| | ↑0.94 | ↑0.60 | ↑1.53 | ↑1.22 | ↑1.20 | ↑0.87 | ↑0.39 | ↑0.17 | ↑0.95 | ↑0.87 | *↑2.04* | *↑1.59* |
| **SEMANTIC SEGMENTATION** | mIoU | mAcc | mIoU | mAcc | mIoU | mAcc | mIoU | mAcc | mIoU | mAcc | mIoU | mAcc |
| WT-Select (Xu et al., 2024) | 28.24 | 36.51 | 35.78 | 44.75 | 26.36 | 34.83 | 29.89 | 38.70 | 22.29 | 30.22 | *28.51* | *37.00* |
| DMAE (Bai et al., 2023) | 27.82 | 36.38 | 32.64 | 41.53 | 28.07 | 37.35 | 31.43 | 40.92 | 24.79 | 33.49 | *28.95* | *37.93* |
| Iso. Pruning (Fang et al., 2024) | 27.34 | 35.00 | 35.89 | 44.73 | 26.89 | 35.12 | 31.44 | 40.05 | 24.12 | 32.00 | *29.14* | *37.38* |
| WAVE (Feng et al., 2025b) | 28.03 | 36.23 | 33.84 | 42.26 | 29.15 | 37.82 | 32.55 | 41.39 | 28.04 | 37.20 | *30.32* | *38.98* |
| SWEET | **29.35** | **37.93** | **38.39** | **47.81** | **31.46** | **40.78** | **37.32** | **47.29** | **28.88** | **38.26** | **33.08** | **42.41** |
| | ↑1.11 | ↑1.42 | ↑2.50 | ↑3.06 | ↑2.31 | ↑2.96 | ↑4.77 | ↑5.90 | ↑0.84 | ↑1.06 | *↑2.76* | *↑3.43* |

of initialization quality. As shown in Table 1, SWEET consistently improves Top-1 accuracy across all model configurations, averaging 1.60 above the strongest baseline. This indicates that the learned weight templates capture multi-scale, category-discriminative features—such as edges, textures, and local shape patterns—that generalize across model depths and widths.

Beyond accuracy, SWEET provides scalable initialization with negligible computational cost. Unlike distillation (e.g., DMAE) or pruning (e.g., Iso. Pruning), which require size-specific optimization or iterative fine-tuning, SWEET leverages pre-trained weight templates through a one-time lightweight weight scaler adaptation, efficiently supporting models of arbitrary sizes. Notably, SWEET surpasses WAVE on image classification, even though WAVE is trained specifically for the task. This demonstrates that self-supervised weight templates effectively capture transferable visual features, and that sharing templates across all components with Tucker decomposition provides a more flexible and scalable model initialization.

### 5.1.2. OBJECT DETECTION

OBJECT DETECTION demands more from initialization due to its reliance on accurate localization and semantic understanding. Despite this, SWEET consistently surpasses scalable initialization baselines across model scales, with stable gains of 2.04 and 1.59 in AP^box and AP^mask, demonstrating its advantage in providing effective, size-agnostic initialization for complex vision tasks.

Unlike classification, object detection requires coordinated representations across the backbone and detection heads. By reconstructing models from a shared weight template, SWEET preserves inter-layer feature consistency, avoiding the misalignment and feature disruption introduced by direct weight selection and enabling reliable scaling without performance degradation. As shown in Fig. 3a, detectors initialized with SWEET achieve more accurate bounding box alignment and reduced localization errors, particularly

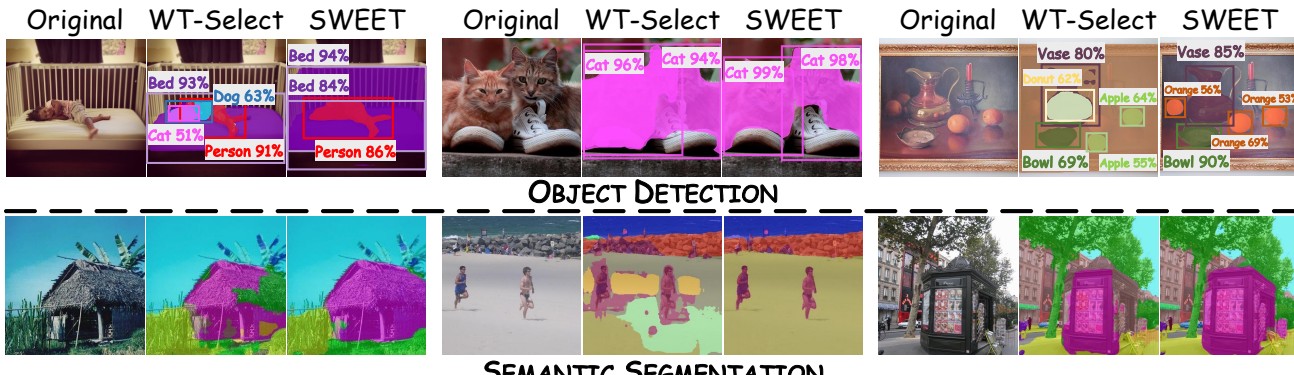

*Figure 3.* Selected visualizations of OBJECT DETECTION and SEMANTIC SEGMENTATION for SWEET-initialized models.

*Table 2.* Scalable initialization performance on IMAGE GENERATION. Models are trained for 50 epochs after initialization.

| | $L_3H_{12}$ | | $L_6H_{12}$ | | $L_6H_6$ | | $L_{12}H_6$ | | $L_{12}H_3$ | | Average | |
|---|---|---|---|---|---|---|---|---|---|---|---|---|
| Para./FLOPs | 33.6M / 11.9G | | 64.3M / 23.8G | | 16.4M / 6.0G | | 31.8M / 11.9G | | 8.1M / 3.0G | | | |
| IMAGE GENERATION | FID | IS | FID | IS | FID | IS | FID | IS | FID | IS | FID | IS |
| WT-Select (Xu et al., 2024) | 31.64 | 21.42 | 17.47 | 33.61 | 35.09 | 18.91 | 21.68 | 27.50 | 47.31 | 13.86 | *30.64* | *23.06* |
| DMAE (Bai et al., 2023) | 29.09 | 23.18 | 15.50 | 37.30 | 33.50 | 19.47 | 21.99 | 26.68 | 44.23 | 14.96 | *28.86* | *24.32* |
| Iso. Pruning (Fang et al., 2024) | 32.29 | 21.23 | 18.68 | 31.53 | 37.60 | 17.44 | 23.83 | 24.92 | 47.06 | 14.09 | *31.89* | *21.84* |
| WAVE (Feng et al., 2025b) | 31.54 | 21.30 | 16.61 | 34.37 | 34.58 | 18.85 | 20.37 | 28.17 | 45.35 | 14.66 | *29.69* | *23.47* |
| SWEET | **27.60** | **24.04** | **14.41** | **38.86** | **31.10** | **20.91** | **19.04** | **30.97** | **41.20** | **15.58** | *26.67* | *26.07* |
| | ↓1.49 | ↑0.87 | ↓1.09 | ↑1.57 | ↓2.40 | ↑1.44 | ↓1.33 | ↑2.80 | ↓3.03 | ↑0.63 | *↓2.19* | *↑1.76* |

for small and medium objects, indicating that the learned weight templates preserve spatial priors and hierarchical feature structures critical for localization.

### 5.1.3. SEMANTIC SEGMENTATION

Semantic segmentation imposes additional challenges for scalable initialization, as it requires dense, pixel-level predictions. Across varying model depths and widths, SWEET consistently outperforms baselines, achieving improvements of 2.76 and 3.43 in mIoU and mAcc.

Fig. 3b further validates this advantage. SWEET-initialized models produce more coherent segmentation maps with sharper boundaries and fewer fragmented regions, indicating that the learned weight templates effectively preserve fine-grained spatial structures and cross-layer consistency.

Collectively, these results show that SWEET provides a unified, scale-robust initialization across vision tasks by leveraging self-supervised weight templates that capture transferable features and preserve structural integrity. In contrast, WAVE, trained only on classification, shows weaker transferability to detection and segmentation, suggesting the benefit of self-supervised templates for cross-task initialization.

### 5.2. Performance on Generative Vision Tasks

Recent studies (Yu et al., 2025; Yun et al., 2025; Lei et al., 2025) have demonstrated that representations learned from discriminative tasks can be leveraged to improve the efficiency of image generation. Building on this, we evaluate self-supervised weight templates on IMAGE GENERATION, a fundamentally more demanding task due to its requirement for holistic content synthesis.

Table 2 shows that SWEET consistently achieves lower FID (↓2.19) and higher IS (↑1.76) than all baselines, reflecting improved visual fidelity and diversity. This advantage arises from weight templates, which preserve rich multi-scale visual priors through regularized constraints, whereas WT-Select and Iso. Pruning, by discarding parameters, can disrupt these priors and compromise cross-scale feature consistency. Moreover, SWEET preserves priors more effectively than classification-trained templates , demonstrating the superior generality of self-supervised representations.

### 5.3. Performance on Downstream Vision Datasets

Beyond standard vision datasets such as ImageNet, we evaluate SWEET on downstream datasets with limited samples and fine-grained semantics, including Oxford Flowers (Nils-

*Table 3.* Performance of models (i.e., $L_6H_6$) on IMAGE CLASSIFICATION with downstream datasets measured by Top-1 Accuracy.

| | Oxford Flower | CUB-200 | Stanford Cars | CIFAR10 | CIFAR100 | Food101 | iNat-2019 | *Average* |
|---|---|---|---|---|---|---|---|---|
| WT-Select (Xu et al., 2024) | 76.65 | 54.99 | 60.98 | 95.21 | 74.10 | 81.95 | 62.67 | *72.36* |
| DMAE (Bai et al., 2023) | **83.18** | 60.84 | 71.41 | 95.82 | 77.36 | 80.78 | 58.84 | *75.46* |
| Iso. Pruning (Fang et al., 2024) | 75.77 | 51.71 | 52.23 | 94.93 | 74.10 | 81.27 | 63.30 | *70.47* |
| WAVE (Feng et al., 2025b) | 80.26 | 56.77 | 57.71 | 93.71 | 75.58 | 82.42 | 63.70 | *72.56* |
| SWEET | 83.12 | **61.51** | **81.03** | **97.03** | **79.36** | **82.50** | **64.22** | ***78.40*** |
| | ↓0.06 | ↑0.67 | ↑9.63 | ↑1.21 | ↑2.00 | ↑0.08 | ↑0.53 | ↑2.01 |

back et al., 2008) and Stanford Cars (Gebru et al., 2017), providing a stringent test of the model's ability to learn general and discriminative visual representations.

As shown in Table 3, SWEET achieves superior performance across most settings, suggesting that self-supervised weight templates capture transferable features that generalize well to these challenging tasks. In contrast, classification-trained templates (e.g., WAVE (Feng et al., 2025b)) and heuristic selection incur larger performance drops, revealing their limited robustness under data scarcity and fine-grained semantic shifts.

### 5.4. Performance on Convolution-based Architectures

Since SWEET operates directly on model weights rather than being tied to any specific architectural design, it naturally extends beyond Transformers to convolutional neural networks (CNNs). This generalization is enabled by a simple structural alignment (Shen et al., 2026), where convolutional kernels are reshaped and concatenated into a unified 2D matrix, making the CNN parameter space compatible with that of Transformers and allowing direct application of the proposed constraint mechanism (see Eq. (5)).

To validate this, we evaluate SWEET on ConvNeXt-v2 (Woo et al., 2023), a modern hierarchical convolutional backbone. Using weight templates learned on ImageNet-1K under the same constraint-based procedure, we conduct experiments on IMAGE CLASSIFICATION. As shown in Table 4, SWEET-initialized ConvNeXt models consistently outperform rule-based and pruning-based baselines across different model scales, demonstrating that the learned structural priors are effectively transferable and enabling a scalable initialization paradigm for convolutional networks.

### 5.5. Ablation and Analysis

#### 5.5.1. EFFECT OF TUCKER-BASED CONSTRAINTS

Constraint-based pre-training regularizes the optimization process to encapsulate knowledge in a decomposable, scale-agnostic form, enabling modular and scalable knowledge extraction with consistent weight reconstruction across model

*Table 4.* Performance on Initializing Convolution-based Architectures. We extend SWEET to ConvNeXt-v2 and evaluate its effectiveness on Image Classification.

| | atto-$L_4$ | femto-$L_6$ | pico-$L_9$ | nano-$L_{12}$ | tiny-$L_{15}$ |
|---|---|---|---|---|---|
| Para./FLOPs | 1.7M | 3.0M | 7.4M | 13.9M | 25.0M |
| | 0.4G | 0.8G | 2.1G | 4.2G | 7.5G |
| WT-Select | 55.92 | 63.13 | 69.17 | 72.23 | 73.87 |
| Iso. Pruning | 50.60 | 57.64 | 65.29 | 69.96 | 72.87 |
| WAVE | 57.20 | 65.38 | 71.69 | 74.90 | 76.57 |
| SWEET | **57.31** | **65.68** | **72.12** | **75.26** | **76.93** |
| | ↑0.11 | ↑0.30 | ↑0.42 | ↑0.36 | ↑0.37 |

*Table 5.* Ablation study on Tucker-based constraints.

| | $L_3H_5$ | $L_4H_4$ | $L_5H_3$ |
|---|---|---|---|
| w/o Constraints | 57.09 | 57.99 | 54.51 |
| Linear | 58.09 | 59.00 | 55.66 |
| Kronecker | 58.83 | 59.47 | 55.88 |
| Tucker (Our) | **58.99** | **59.71** | **56.70** |

*Table 6.* Ablation study on width-wise stochastic scaling (w/o Stoch. Scal.) and architectural enhancements (w/o Arch. Enh.), including SwiGLU, RMSNorm, and RoPE.

| | $L_3H_3$ | $L_3H_4$ | $L_3H_5$ |
|---|---|---|---|
| w/o Stoch. Scal. | 48.63 | 53.92 | 58.65 |
| WT-Select | 48.65 | 53.64 | 57.09 |
| w/o Arch. Enh. | 49.03 | 54.65 | 57.80 |
| SWEET | **49.57** | **55.21** | **58.80** |

sizes. In contrast, as shown in Table 5, unconstrained pre-training produces tightly coupled, scale-specific weights that generalize poorly to unseen widths or depths.

Compared with linear and Kronecker-based constraints, Tucker-based constraints provide greater flexibility by disentangling knowledge across layer and width dimensions. Linear factorization is limited to a single subspace, while Kronecker decomposition enforces rigid multiplicative structure, both restricting scalability. In contrast, Tucker-based

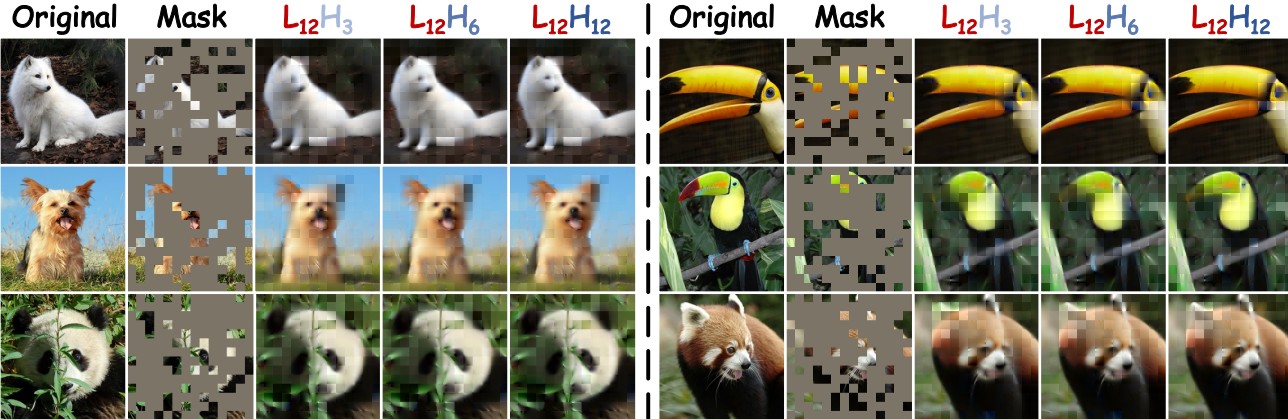

*Figure 4.* Reconstructions of ImageNet validation images using SWEET pre-trained weight templates with a masking ratio of 75%.

constraints, combined with width-wise stochastic scaling, balance expressiveness and regularization to enable size-agnostic and robust initialization across model scales.

### 5.5.2. EFFECT OF WIDTH-WISE STOCHASTIC SCALING

Width-wise stochastic scaling regularizes knowledge along width-related dimensions by applying structured dropout to weight scalers during pre-training, forcing the model to reorganize information across the width dimension. This stochastic perturbation encourages low-index components of templates to capture width-invariant representations.

As shown in Table 6, width-wise stochastic scaling substantially improves the adaptability of learned templates across models of varying widths, consistently outperforming variants without scaling and yielding higher classification accuracy. We further examine the effect of general architectural enhancements on the structure of weight templates (Sec. 3.3). Integrating components such as SwiGLU, RMSNorm, and RoPE enhances template expressiveness and stability, promoting consistent feature encoding and improving performance across diverse model configurations.

### 5.5.3. VISUALIZATION OF MAE RECONSTRUCTION

We analyze the learned self-supervised weight templates via MAE-based per-element reconstruction under a 75% masking ratio, as illustrated in Fig. 4. Across model scales, the templates consistently reconstruct masked components, indicating that core structural and semantic information is compactly encoded in the unmasked parameters. Moreover, reconstruction quality remains stable as model size varies, suggesting that the learned templates capture scale-agnostic and reusable representations rather than architecture-specific patterns. Together, these results provide qualitative evidence that constraint-based pre-training yields coherent and decomposable weight templates, enabling robust reconstruction and transfer across diverse model configurations.

## 6. Conclusion

We present SWEET, a constraint-based pre-training framework for learning size-agnostic weight templates in a self-supervised manner. By combining Tucker-based constraints with width-wise stochastic scaling, SWEET encodes transferable visual knowledge into reusable templates, enabling scalable and task-agnostic model initialization. Extensive experiments on CLASSIFICATION, DETECTION, SEGMENTATION, and GENERATION show that SWEET consistently outperforms existing scalable initialization methods while maintaining robust performance across diverse model scales.

## Impact Statement

The broader impact of our work lies in how SWEET redefines vision model initialization through the learning of size-agnostic, self-supervised weight templates that enable efficient initialization across tasks and scales. By offering a unified, scalable, and generalizable initialization paradigm, SWEET has the potential to accelerate research on AI model scaling, improve performance in low-data and specialized settings, and promote more sustainable and flexible industrial AI applications.

## Acknowledgement

We sincerely appreciate Freepik for contributing to the figure design. This research was supported by the Jiangsu Science Foundation (BG2024036, BK20243012), the National Natural Science Foundation of China (625B2045, 62125602, U24A20324, 92464301, 62306073), the New Cornerstone Science Foundation through the XPLORER PRIZE, the Fundamental Research Funds for the Central Universities (2242025K30024), and SEU Innovation Capability Enhancement Plan for Doctoral Students (CXJH_SEU 26023).

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

# A. Training Details

## A.1. Details of Tucker-based Pre-training

Algorithm 1 provides a detailed description of the Tucker-based pre-training procedure used to learn weight templates.

---

**Algorithm 1** Tucker-constrained Pre-training for Learning Weight Templates

---

**Input**: An $L$-layer ViT with parameters $\theta$, Training dataset $\{(x^{(i)})\}_{i=1}^m$, Number of training epochs $N_{\text{ep}}$, Batch size $B$, Learning rate $\alpha$

**Output**: Weight Templates $\mathcal{G}$, Weight Scalers $(U, V, X)$

1: Concatenate all learnable weight matrices in $\theta$ across $L$ layers into a unified weight tensor $\mathcal{W}$
2: Random initialize Weight Templates $\mathcal{G}$ and Weight Scalers $(U, V, X)$
3: Construct the instantiated weight tensor $\mathcal{W}$ via Tucker composition (Eq. (5))
4: **for** $ep = 1$ to $N_{\text{ep}}$ **do**
5:     **for** each mini-batch $\{(x_i)\}_{i=1}^B$ **do**
6:         Reconstruct the weight tensor $\mathcal{W}$ from $(\mathcal{G}, U, V, X)$ using Eq. (5)
7:         Perform forward propagation to obtain predictions $\hat{x}_i = f_\theta(x_i)$
8:         Compute the mini-batch loss $\mathcal{L}_{\text{batch}} = \frac{1}{B} \sum_{i=1}^B \mathcal{L}(\hat{x}_i, x_i)$ according to Eq. (1)
9:         Backpropagate $\mathcal{L}_{\text{batch}}$ to compute gradients with respect to $\mathcal{G}, U, V$ and $X$
10:        Update the Tucker parameters via gradient descent:

$$\mathcal{G} \leftarrow \mathcal{G} - \alpha \nabla_{\mathcal{G}} \mathcal{L}_{\text{batch}}, \quad U \leftarrow U - \alpha \nabla_U \mathcal{L}_{\text{batch}}, \quad V \leftarrow V - \alpha \nabla_V \mathcal{L}_{\text{batch}}, \quad X \leftarrow X - \alpha \nabla_X \mathcal{L}_{\text{batch}}$$

11:     **end for**
12: **end for**

---

## A.2. Details of Datasets

Table 7 provides an overview of the datasets used in our experiments. ImageNet-1K (Deng et al., 2009) is utilized for constraint-based pre-training to learn self-supervised weight templates and is further adopted for IMAGE CLASSIFICATION and IMAGE GENERATION. ADE20K (Zhou et al., 2019) is used for SEMANTIC SEGMENTATION, while COCO (Lin et al., 2014) is employed for OBJECT DETECTION. In addition, we evaluate transferability on eight downstream IMAGE CLASSIFICATION datasets to examine the generalization of the learned weight templates across diverse data distributions.

## A.3. Hyper-parameters

We provide an overview of the hyper-parameter settings employed across all experiments. Table 8 details the configurations for constraint-based pre-training of self-supervised weight templates, as well as for tasks including IMAGE CLASSIFICATION, SEMANTIC SEGMENTATION, OBJECT DETECTION, and IMAGE GENERATION. Table 9 specifies task-specific hyper-parameters for downstream IMAGE CLASSIFICATION on eight datasets, including learning rates and training epochs.

# B. Additional Results

Fig. 5 presents additional visualizations of MAE reconstructions obtained using SWEET pre-trained weight templates, providing further qualitative evidence of the reconstruction capability under high masking ratios.

*Table 7.* Characteristics of datasets.

| Dataset | Classes | Total | Training | Testing |
|---|---|---|---|---|
| **ImageNet** (Deng et al., 2009) | 1,000 | 1,331,167 | 1,281,167 | 50,000 |
| **ADE20K** (Zhou et al., 2019) | 150 | 22,210 | 20,210 | 2,000 |
| **COCO** (Lin et al., 2014) | 80 | 123,287 | 118,287 | 5,000 |
| **Oxford Flowers** (Nilsback et al., 2008) | 102 | 8,189 | 2,040 | 6,149 |
| **CUB-200-2011** (Wah et al., 2011) | 200 | 11,788 | 5,994 | 5,794 |
| **Stanford Cars** (Gebru et al., 2017) | 196 | 16,185 | 8,144 | 8,041 |
| **CIFAR10** | 10 | 60,000 | 50,000 | 10,000 |
| **CIFAR100** | 100 | 60,000 | 50,000 | 10,000 |
| **Food101** (Bossard et al., 2014) | 101 | 101,000 | 75,750 | 25,250 |
| **iNat-2019** (Tan et al., 2019) | 1,010 | 268,243 | 265,213 | 3,030 |

*Table 8.* Hyper-parameters for SWEET in constraint-based pre-training and downstream tasks, including IMAGE CLASSIFICATION, SEMANTIC SEGMENTATION, OBJECT DETECTION, and IMAGE GENERATION.

| Training Settings | Pre-training | CLASSIFICATION | DETECTION | SEGMENTATION | GENERATION |
|---|---|---|---|---|---|
| **Image Size** | 224 | 224 | 1024 | 512 | 64 |
| **Patch Size** | 16 | 16 | 16 | 16 | 4 |
| **Optimizer** | AdamW | AdamW | AdamW | AdamW | AdamW |
| **Base Learning Rate** | 1.5e-4 | 5e-4 | 1e-4 | 1e-4 | 5e-5 |
| **Warmup Learning Rate** | 0 | 0 | 1e-7 | 0 | 0 |
| **Weight Decay** | 0.05 | 0.05 | 0.1 | 0.05 | 0 |
| **Optimizer Momentum** | 0.9 | 0.9 | 0.9 | 0.9 | 0.9 |
| **Batch Size** | 1024 | 256 | 16 | 16 | 256 |
| **Training Epochs** | 450 | 30 | 30 | 126 | 50 |
| **Learning Rate Scheduler** | Cosine Decay | Cosine Decay | MultiStep Decay | Poly Decay | Constant |
| **Drop Path** | 0.3 | 0 | 0.1 | 0.1 | 0 |
| **Warmup Epochs** | 22 | 0 | 0.03 | 1.18 | 5 |
| **Random Erase** | — | 0.25 | — | — | — |
| **Label Smoothing** | — | 0.1 | — | — | — |
| **Time Sampler** | — | — | — | — | $\sim \mathcal{N}(-0.8, 0.8^2)$ |
| **Noise Scale** | — | — | — | — | 0.25 |
| **Sample Solver** | — | — | — | — | Heun |
| **Sample Step** | — | — | — | — | 50 |
| **Cfg Scale** | — | — | — | — | 2.9 |

*Table 9.* Hyper-parameters for IMAGE CLASSIFICATION on Downstream Datasets.

| Dataset | Oxford Flowers | CUB-200-2011 | Stanford Cars | CIFAR10 | CIFAR100 | Food101 | iNat-2019 |
|---|---|---|---|---|---|---|---|
| **Batch Size** | 512 | 512 | 512 | 512 | 512 | 512 | 512 |
| **Epoch** | 300 | 300 | 300 | 300 | 300 | 300 | 50 |
| **Learning Rate** | 3e-4 | 3e-4 | 3e-4 | 5e-4 | 5e-4 | 5e-4 | 5e-4 |
| **Drop Last** | False | False | False | True | True | True | True |
| **Warmup Epochs** | 0 | 0 | 0 | 0 | 0 | 0 | 0 |
| **Droppath Rate** | 0 | 0.1 | 0.1 | 0.1 | 0.1 | 0.1 | 0.1 |
| **Random Erase** | 0.25 | 0.25 | 0.25 | 0.25 | 0.25 | 0.25 | 0.25 |
| **Mixup** | 0 | 0 | 0 | 0 | 0 | 0 | 0 |
| **Cutmix** | 0 | 0 | 0 | 0 | 0 | 0 | 0 |
| **Scheduler** | Cosine | Cosine | Cosine | Cosine | Cosine | Cosine | Cosine |
| **Optimizer** | AdamW | AdamW | AdamW | AdamW | AdamW | AdamW | AdamW |
| **Auto Augment** | | | rand-m9-mstd0.5-inc1 | | | | |

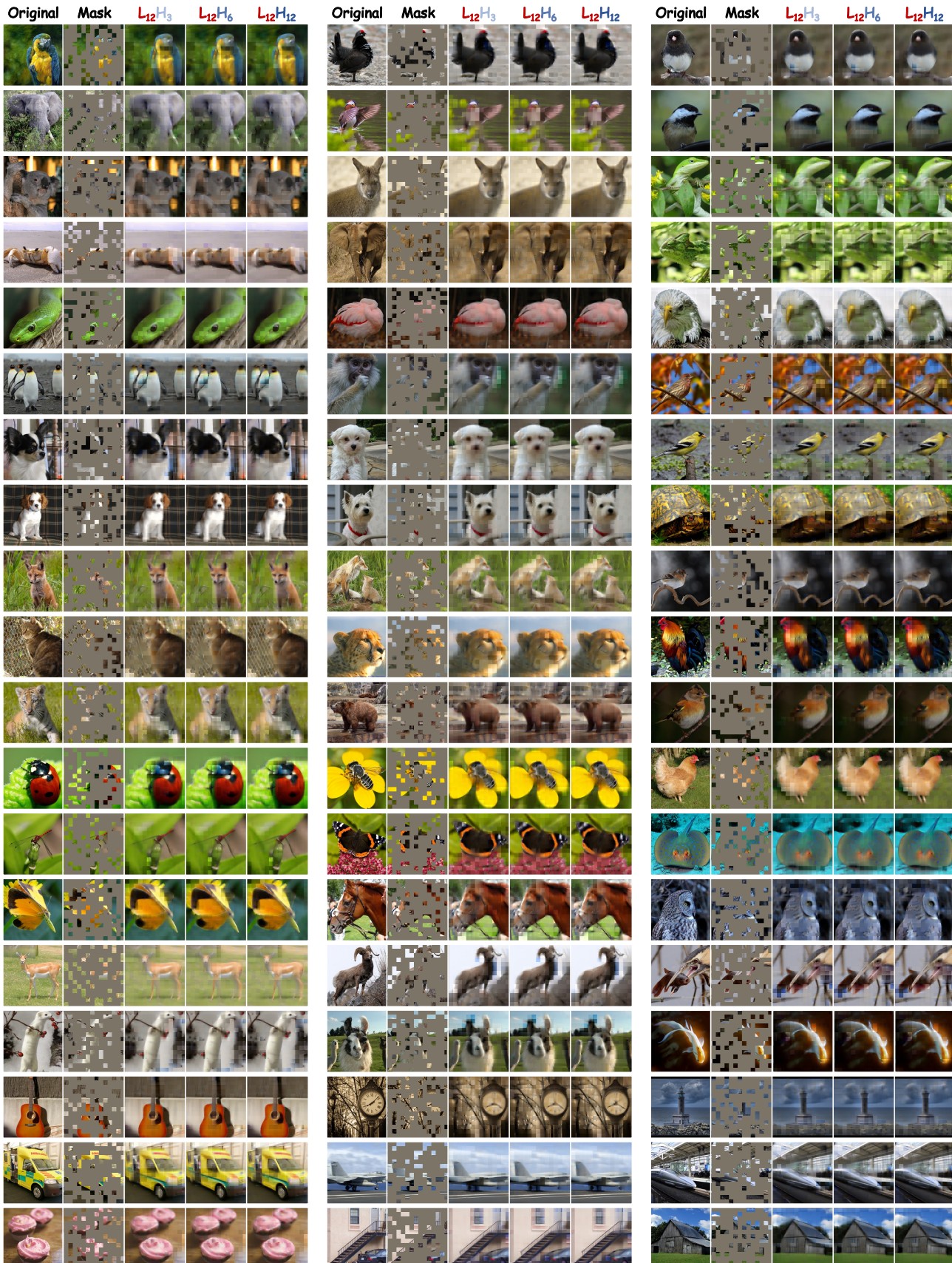

*Figure 5.* Additional visualizations of MAE reconstructions with SWEET pre-trained weight templates under a 75% masking ratio.

