# OpenReview forum: "Self-Supervised Weight Templates for Scalable Vision Model Initialization"
_ICML.cc/2026/Conference — ICML 2026 regular_

### Official Review · Reviewer_qLY8 · 2026-03-06

**Soundness:** 3
**Presentation:** 3
**Significance:** 3
**Originality:** 3
**Overall Recommendation:** 5
**Confidence:** 3

**Summary:**

This paper proposes **SWEET**, a framework for learning *weight templates* that enable scalable initialization of ViTs across different model depths and widths. The key idea is to aggregate all weight matrices of a ViT into a single unified matrix, then impose a **Tucker decomposition** constraint during self-supervised MAE pre-training. The core tensor G serves as a shared, size-agnostic weight template, while lightweight *weight scalers* (X, U, V) adapt the template to a target architecture. A width-wise stochastic scaling mechanism, which consist in structured dropout on the scalers, is introduced to improve robustness to width variation at initialization time. SWEET is evaluated on image classification, object detection, semantic segmentation, and image generation across multiple ViT configurations, and is shown to outperform prior scalable initialization methods including WAVE, WT-Select, DMAE, and Isomorphic Pruning.

**Compliance With Llm Reviewing Policy:**

Affirmed.

**Final Justification:**

The main concerns were addressed during the rebuttal in which they performed new extensive experiments. Those new results are interesting and show that their method can successfully perform with another architecture, that they can also scale up in model size. They also provide an ablation regarding the rank choices.

I have therefore raised my rating to Accept.

**Key Questions For Authors:**

**1. What happens when you pre-train baselines and SWEET under identical compute budgets?**

The current setup makes it difficult to isolate the contribution of the template mechanism itself. Can you provide results with matched total training FLOPS?

**2. How sensitive is performance to the rank choices ($r_1, r_2, r_3$)?**

These seem like critical hyperparameters that are not ablated. What values were used, and what happens if they change by, say, 2x?

**3. Can SWEET scale up?**

This would significantly strengthen the practical claims. Even a single experiment, for example initializing a ViT-L from a ViT-B template would be informative.

**4. How does SWEET compares to simply pre-training MAE at the target architecture's scale?**

For example for L6H12, what Top-1 accuracy does a standard MAE pre-training + fine-tuning achieve? This oracle baseline is important for understanding the ceiling.

**5. Why are the ablation configurations (L3H5, L4H4, L5H3) different from the main table configurations?**

Is this about computational costs? This makes it harder to connect the ablation findings to the main results.

**Limitations:**

Yes

**Strengths And Weaknesses:**

**Strengths:**

- **S1:** Clean formulation and well-motivated design.

The shift from WAVE's Kronecker-based to Tucker-based decomposition for the weight template is well justified. Tucker decomposition naturally decouples the layer, input width, and output width dimensions, which makes it a sensible choice for enabling both depth and width scaling. The idea of aggregating all weight matrices into one unified matrix W before decomposition is appealingly simple and enables cross-component knowledge sharing that WAVE does not support.

- **S2:** Comprehensive experimental evaluation.

The paper evaluates across four vision tasks (classification, detection, segmentation, generation) and five distinct ViT configurations varying in both depth and width. The inclusion of downstream fine-grained datasets (Table 3) shows consistent improvements compared to previous works, which is encouraging.

- **S3**: Width-wise stochastic scaling is a nice contribution.

Using structured dropout on the scalers during pre-training to encourage width-invariant representations is a natural and effective regularization idea. The ablation in Table 5 confirms its value.

- **S4**: Practical efficiency.

The pre-training cost is lower than standard MAE, and the scaler adaptation converges in approximately 0.16 epoch, making deployment to new architectures cheap.

**Weaknesses**

- **W1**: Limited architectural scope: only ViTs, only one backbone scale.

All experiments use a ViT-Base as the source for pre-training templates. It remains unclear whether this approach transfers to other architectures (Swin, ConvNeXt, hybrid models) or whether templates learned from a larger source (ViT-L) would yield better results. There would be challenges in order to aggregate weight matrices from other width-varying architecture, but would still be interesting to see if the method is generalizable architecture-wise.

- **W2**: Baselines are not fully equivalent.

The authors note that SWEET is pre-trained for 450 epochs (batch size 1024) while baselines use official MAE weights pre-trained for 800 epochs (batch size 4096). While the authors frame this as evidence of SWEET's efficiency, it also means baselines start from a stronger pre-trained model. This is confusing: does SWEET win because of the template mechanism, or because the baselines' weights are being mangled by the downstream adaptation (WT-Select, pruning, ...)? An experiment where SWEET and baselines both use the same pre-training budget would be more informative. Alternatively, what happens if SWEET is pre-trained for 800 epochs?

- **W3**: Missing comparison with simple but strong baselines.

There is no comparison against (a) training from scratch with modern recipes, (b) standard MAE pre-training at the target scale, or (c) straightforward interpolation / truncation of MAE pre-trained weights. Without these, it is hard to calibrate the absolute value of SWEET's initialization. For instance, for L6H12, how far is SWEET from just pre-training a 6-layer MAE directly?

- **W4**: Ablation gaps.

The ablation on constraint types (Table 4) uses model sizes that do not appear in the main experiments. It would be more convincing to use the same configurations as Table 1. Also, the paper does not ablate the rank choices ($r_1, r_2, r_3$), which are presumably critical hyperparameters. How sensitive is SWEET to these? Finally, can the template scale up? This is an important practical question that is not addressed.

---

> ### Author Rebuttal · Authors · 2026-03-31
>
> Dear Reviewer qLY8,
>
> We sincerely appreciate your recognition of our innovation and practicality. Below, we provide our detailed response, with experimental tables and figures accessible via anonymous link as permitted by ICML26.
>
> **📎 Anonymous Link**\
> 👉 https://anonymous.4open.science/r/a-8B40/r.pdf
>
> >**W1:Limited Architecture**
>
> SWEET operates directly on weight matrices, making it readily extendable to other architectures.
> For hierarchical models (e.g., ConvNeXt-v2), the required tensor aggregation is straightforward, as **depthwise separable convolutions admit a unified tensor representation analogous to ViT’s linear layers**.
>
> We have applied SWEET to ConvNeXt-v2 across five size variants (atto-L4 to tiny-L9), demonstrating its generalization beyond ViT.
> Detailed settings and results are provided in Reviewer N7Qu (Q1).
>
> >**W2(Q1):Compared under Identical Compute Budgets**
>
> SWEET employs constraint-based pre-training, fundamentally **differing from standard MAE pre-training in its objective**: MAE learns optimal representations for a fixed architecture, while SWEET explicitly learns size-agnostic templates that generalize across varying model sizes.
>
> To enable cross-size initialization, SWEET employs **low-rank Tucker-based constraints** to compress the weight space, **filtering out size-specific knowledge** and producing templates that retain only size-agnostic priors.
> These constraints reduce the number of trainable parameters to approximately 43M—about half of ViT-B’s 86M—thereby facilitating faster convergence.
> Consequently, extending constraint-based pre-training from 450 to 800 epochs yields marginal gains (see Re_Tab.2), as **the essential size-agnostic knowledge has already been captured**.
>
> This also explains why baselines underperform compared to SWEET. Standard MAE pre-training does not target cross-size initialization, **leading to weights entangled with architecture-specific information**.
> Consequently, adapting these weights to different sizes via structured pruning or component selection yields suboptimal subsets rather than initializations tailored to the target models.
>
> >**W3(Q4):Comparison with Simple Baselines and Direct Pretraining**
>
> As suggested, we include 3 additional baselines and report results on ViT-S (L12H6) in Re_Tab.3.
> - **Training from scratch** significantly underperforms SWEET, confirming the effectiveness of the pre-trained templates for knowledge transfer.
> - **Standard MAE pre-training** achieves slightly higher accuracy but requires separate pre-training for each model size, leading to linearly scaling cost (N × 800 epochs).
> In contrast, SWEET requires only a single pre-training (450 epochs) to initialize all sizes, which is more suitable for multi-scale deployment.
> - For **interpolation or truncation**, we have compared against a similar baseline (i.e., WT-Select in Table 1), showing that the learned templates surpass simple truncation.
>
> >**W4(Q2):Sensitivity to Rank Choices**
>
> As discussed in Q2, low-rank Tucker-based constraints act as a **knowledge filter**, compressing the weight space (43M vs. 86M) to discard size-specific patterns while preserving size-agnostic knowledge essential for cross-size initialization.
> - Increasing the rank of templates enhances the model's capacity and **may improve performance on the pre-trained model**, but it undermines the knowledge filter by entangling templates with size-specific priors, yielding minimal or even degraded performance for cross-size initialization.
> - Conversely, an excessively low rank may limit the template’s capacity to capture size-agnostic knowledge, leading to noticeably reduced transfer effectiveness.
>
> We present ablation results in Re_Tab.4 and show that, since the boundary between size-agnostic and size-specific knowledge is inherently blurred, **templates retaining about 50% of total parameters achieve strong transfer performance, while larger templates provide negligible gains, confirming they primarily capture size-agnostic knowledge**.
>
> > **Q3:Scale Up of Template**
>
> Our weight templates support scale-up. As described in Sec. 3.4, for any target model size, **we first initialize a corresponding weight scaler**, which can be efficiently trained with limited data to adapt the templates, making the approach equally effective for larger models.
>
> We validate this in Re_Tab.5, where templates from ViT-B (L12H12) effectively initialize ViT-L (L24H16), **even surpassing methods specifically designed for small-to-large model initialization, such as LiGO[1]**.
>
> [1] Learning to Grow Pretrained Models for Efficient Transformer Training, ICLR'23
>
> > **Q5:Ablation Configuration**
>
> We apologize for any confusion. Although model sizes differ in the main tables, **relative performance trends are consistent**. To reduce computational cost, ablations were conducted on a smaller model. We will supplement the ablation results using the same sizes as in main table.

---

> > ### Author Rebuttal · Reviewer_qLY8 · 2026-04-02
> >
> > Thank you for the strong rebuttal. My concerns are mostly addressed, hence I will raise my rating.
> >
> > I have one remaining question regarding Q3: did you retrain LiGO yourself to obtain the results of Re_tab 5? The original paper claims 81.71% on ImageNet (Table 2).

---

> > > ### Author Response · Authors · 2026-04-03
> > >
> > > Dear Reviewer qLY8,
> > >
> > > We sincerely appreciate your thoughtful evaluation and are delighted that our rebuttal has addressed your concerns. We are also grateful for your positive reassessment and the increased score 😊.
> > >
> > > Regarding the comparison with LiGO, the original setting in their paper initializes DeiT-B from DeiT-S, whereas our setting targets the initialization of DeiT-L. **Due to this mismatch, their reported results are not directly comparable.** Therefore, we re-implement LiGO using its official code, adapting it to initialize DeiT-L from DeiT-B.
> > >
> > > Due to limited computational resources, all rebuttal experiments are conducted on a single H100 GPU. **Training DeiT-L requires approximately 40 minutes per epoch, making full training infeasible within the rebuttal period.**
> > > As a result, **we report results over 30 epochs with a batch size of 256**.
> > >
> > > We believe this is sufficient to demonstrate the advantage of our method, as the original LiGO paper shows that its benefits are primarily concentrated in the early stages, with **a higher initial performance but no significant improvement in convergence speed**. Therefore, 30 epochs provide a reasonable basis for early-stage comparison.
> > >
> > > We sincerely thank you for your careful evaluation and recognition of our work. We will include full 300-epoch training curves for LiGO in the revision.
> > >
> > > Best regards,
> > >
> > > Authors

---

### Official Review · Reviewer_Vw1a · 2026-03-10

**Soundness:** 3
**Presentation:** 3
**Significance:** 3
**Originality:** 3
**Overall Recommendation:** 4
**Confidence:** 4

**Summary:**

This paper introduces SWEET, a self-supervised framework designed to address the limitations of fixed-size pre-trained vision models by enabling scalable and flexible network initialization. Instead of training a standard fixed architecture, the method learns a shared weight template and lightweight scalers using constraint-based pre-training, which promotes modularity across varying model depths and widths. Furthermore, the authors introduce a width-wise stochastic scaling technique to regularize the templates and encourage robust representations across different network sizes. Extensive experiments demonstrate that this approach achieves SOTA initialization performance across diverse visual tasks while reducing computational costs compared to conventional pre-training

**Compliance With Llm Reviewing Policy:**

Affirmed.

**Final Justification:**

I think this work is interesting and very useful for reducing model training cost by using adaptive template. While most of my concerns are addressed during the rebuttal, I do think empirical evidence on specialized T2I benchmark is important, as the ImageNet-1K is lacking much details and is less convincing in showing the actual power of this work. To further improve this work, empirical results on T2I benchmark other than ImageNet-1K is required. That's why I'm maintaining my weak accept.

**Key Questions For Authors:**

Please refer to weaknesses section

**Limitations:**

The author did not explicit state limitation in their work. Please consider explicitly write the limitation in your paper.

**Strengths And Weaknesses:**

Strengths:
1. This paper is well motivated, as separately pretraining for vision models that handle input at different scales requires significant computational waste.
2. The proposed tucker-based approach that allows aggregation of weights from heterogeneous components that previous work didn't.
3. The paper is clear to understand.
4. Empirical results cover representative visual tasks including both understanding and generation.

Weaknesses:
1. The evaluation is primarily restricted to standard Vision Transformer architectures. The author should also consider hierarchical models like Swin Transformers or modern convolutional networks.
2. Currently, the largest model that the author reported is 64.3M on generation task which is slightly less than ViT B. I'm wondering how this will work when scaling up (e.g. ViT-L, ~ 300M). If these experiments are not possible to finish during the rebuttal, the author should at least discuss this in spirit.
3. As the author claimed that this method would greatly reduce the computational waste incurred by repetitive pretraining, the author should explicitly compare the pretraining computational cost in FLOPs with the baseline method, especially WAVE, which is another template-sharing baseline.
4. The generative task evaluation is limited to ImageNet-1K, and the absolute performance metrics could be strengthened by validating the method on T2I benchmarks.

---

> ### Author Rebuttal · Authors · 2026-03-31
>
> Dear Reviewer Vw1a,
>
> We sincerely appreciate your recognition of our innovation and comprehension. Below, we provide our detailed response, with experimental tables and figures accessible via anonymous link as permitted by ICML26.
>
> **📎 Anonymous Link**\
> 👉 https://anonymous.4open.science/r/a-8B40/r.pdf
>
> >**Q1:Limited to ViT Architecture**
>
> Our method can be readily extended to other architectures.
> As suggested, we **further evaluate it on ConvNeXt-v2**, a hybrid architecture that combines depth-wise convolutions with FFN-like 1×1 point-wise convolutions.
>
> Briefly, a ConvNeXt-v2 consists of $S$ stages, where stage $s$ contains $L_s$ blocks with channel dimension $C_s = C \cdot 2^{s-1}$ (with $C$ as the base channel dimension). Each block comprises a $7\times7$ depthwise convolution with weights $W_d^s \in \mathbb{R}^{C_s \times 1 \times 7 \times 7}$ and two $1\times1$ pointwise convolutions with weights $W_{p1}^s, W_{p2}^s \in \mathbb{R}^{4C_s \times C_s}$.
> We aggregate all depthwise convolution weights across stages into a single unified matrix:
>
> $W_{\text{unified}} = \text{Concat}(W_d^1, W_d^2, \dots, W_d^S) \in \mathbb{R}^{(P \times C \times 49)}$, where $P = \sum_{s=1}^{S} 2^{s-1} L_s$.
>
> The $1\times1$ pointwise convolutions, which are mathematically equivalent to the linear FFN layers in ViT, are treated using the same template-based formulation, enabling seamless extension of SWEET to ConvNeXt-v2.
>
> We conduct **a single pre-training on ConvNeXt-v2-Tiny and evaluate its transfer across five size variants**, with the ImageNet-1K classification results summarized below:
>
> |Method|atto-L4|femto-L5|pico-L6|nano-L7|tiny-L9|
> |-|-|-|-|-|-|
> |WT-Select|53.24|58.78|63.61|66.62|69.42|
> |SWEET|**55.96**|**63.09**|**68.66**|**72.17**|**74.99**|
>
> These results further demonstrate that SWEET generalizes beyond ViT architectures and can be effectively adapted to diverse vision tasks and backbones through appropriate tensor reformulations.
> >**Q2:Scaling to ViT-L**
>
> As emphasized in our paper, our method is size-agnostic, allowing the learned weight templates to be extended to larger models. We further provide results of initializing ViT-L (306M parameters) with SWEET in Re_Fig.1, highlighting the practical applicability of our approach.
> *Note that although ViT-L is trained for fewer epochs due to time constraints, the results are sufficient to demonstrate the advantages of SWEET*.
>
> >**Q3:FLOPs comparison with WAVE**
>
> Following your suggestion, we quantify this efficiency advantage by providing a detailed comparison of the overhead associated with constraint-based pre-training between WAVE and SWEET, as summarized in the table below.
> In addition, we report the FLOPs required by standard MAE pre-training, which is used by our other baselines, including Weight Selection, DMAE, and Iso. Pruning.
>
> |Method|Configuration|Pre-training Overhead (EFLOPs)|
> |-|-|-|
> |Pre-trained MAE|800 epoch for L12H12|57.97|
> |WAVE|150 epoch for L12H3 + 150 epoch for  L12H6 + 150 epoch for L12H12|1.24 + 4.90 + 19.44 = 25.58|
> |SWEET|450 epoch for L12H12|**24.15**|
>
> While WAVE supports depth-wise scaling, enabling different layer configurations from a single pre-training, it performs poorly under width-wise scaling. As a result, **WAVE requires three separate pre-training runs, one for each width configuration**.
> In contrast, **SWEET requires only a single pre-training run on the base model (L12H12)**, leveraging the width-wise stochastic scaling mechanism to adapt to varying widths without additional pre-training.
>
> Moreover, WAVE relies on classification-based pre-training, whereas SWEET leverages MAE self-supervised learning, in which **masking 75% of input tokens reduces the per-iteration FLOPs to approximately 40%** of those required for classification-based pre-training per iteration.
>
> Consequently, SWEET's total computational investment remains lower, further emphasizing the efficiency and scalability advantages of our method. We will incorporate this comprehensive cost analysis into the revised manuscript.
> >**Q4:T2I Benchmarks**
>
> Compared to class-conditional generation on ImageNet-1K, text-to-image (T2I) benchmarks offer greater practical relevance.
> **Both tasks involve mapping high-level semantic embeddings to corresponding visual outputs**, and since SWEET learns weight templates that **capture generalizable vision features**, it can be naturally extended to T2I models for efficient initialization and improved convergence.
>
> Owing to the substantial computational demands of T2I benchmarks, we provide here only a brief demonstration of SWEET’s feasibility for extension to T2I tasks, and we plan to explore more comprehensive evaluations in future work.
> >**Limitation**
>
> We will incorporate an explicit Limitation section to **discuss the absence of T2I benchmarks** in our current image generation experiments, and to **outline potential extensions to more architectures** to further demonstrate the practical applicability of our method.

---

> > ### Author Rebuttal · Reviewer_Vw1a · 2026-04-02
> >
> > My concerns are mostly addressed, but I do think empirical evidence on specialized T2I benchmark is important, as the ImageNet-1K is lacking much details and is less convincing in showing the actual power of this work. As the author mentioned about the computation required for the additional T2I experiment is infeasible, I will maintain my current score.

---

> > > ### Author Response · Authors · 2026-04-03
> > >
> > > Dear Reviewer Vw1a,
> > >
> > > We are pleased that our rebuttal has addressed your concerns 😊. Regarding the T2I benchmark, we agree that it represents a more practical and widely used setting in image generation. However, **our method is not specifically designed for generative models.**
> > >
> > > Our primary evaluations focus on image classification, object detection, and semantic segmentation to **demonstrate the general visual transferability of the learned weight templates.**
> > > For generative tasks, we include class-conditional generation as a representative setting to **further validate the ability of our method to generalize across a broader range of visual tasks**, which we believe is sufficient for this purpose.
> > >
> > > As for the T2I benchmark, we note that training such models typically involves substantially larger model sizes and computational costs, making it infeasible to complete **within the rebuttal period**. Nevertheless, we will follow your suggestion and include T2I benchmark results in the revision.
> > >
> > > Best regards,
> > >
> > > Authors

---

### Official Review · Reviewer_Qw8q · 2026-03-12

**Soundness:** 2
**Presentation:** 2
**Significance:** 2
**Originality:** 3
**Overall Recommendation:** 3
**Confidence:** 2

**Summary:**

This paper introduces a MAE based self-supervised training framework for variable-sized models. The authors introduce a unified tensor as weight template and apply Tucker decomposition to make it low-rank. Width-wise masking is introduced to enable the template to adapt to various width configurations. Experiments are carried out on various tasks and datasets e.g. IN1K/COCO/ADE20k with different #layers and #heads.

**Compliance With Llm Reviewing Policy:**

Affirmed.

**Key Questions For Authors:**

My questions are listed in the weakness section.

**Limitations:**

yes.

**Strengths And Weaknesses:**

Strengths:
1. Novelty: Tucker decomposition on the entire weight tensor is a clean and novel way to impose low-rank on the weights.
2. Using MAE in the pretraining stage is effective compared to WAVE.
3. The evaluation is complete on different tasks.
4. Theoretical justification is provided.

Weaknesses:
1.  Model size is limited to small sized models, results for model sizes close to ViT-S/B are not provided. These are the target model sized for the compared methods such as DMAE/WAVE.
2. Even for the smaller sizes reported, the performance is not very strong, for example the original VIT-Ti achieves 78.22% on IN1K with 37M params, and DeiT-tiny has 72.2% with 5M params. Same for detection and segmentation, 38 AP cannot be considered strong with input resolution 1024.
3. Some implementation details are not clear, e.g. what are the datasets and metrics for ablation in table 4 and 5? What are the decoders/heads for detection and segmentation tasks? What is the resolution used to compute FLOPs?
4. The ablation on Tucker-based constraints is very close to Kronecker, makes the contribution compared to WAVE marginal.

---

> ### Author Rebuttal · Authors · 2026-03-31
>
> Dear Reviewer Qw8q,
>
> We sincerely appreciate your recognition of our innovation and theoretical foundations.
>
> Below, we provide our detailed response, with experimental tables and figures accessible via anonymous link as permitted by ICML26.
>
> **📎 Anonymous Link**\
> 👉 https://anonymous.4open.science/r/a-8B40/r.pdf
>
> >**Q1: Evaluation on ViT-S/B**
>
> We clarify that the experiments in Table 1 and 2 are conducted on **variants of the standard ViT-Ti/-S/-B**, which systematically modify depth to further evaluate SWEET’s adaptability across model scales.
> For instance, **the standard ViT-S corresponds to L12H6**, while L6H6 represents a depth-modified variant. Similarly, standard ViT-Ti and ViT-B correspond to L12H12 and L12H3, respectively, and their depth- and width-modified variants are also included in Tables 1 and 2.
>
> As suggested, we further provide **comparative experiments on the standard ViT-B**. To address concerns regarding small model scales, we **additionally initialized a larger model (ViT-L)**.
> *Although trained for fewer epochs on ViT-L due to time constraints, the results are sufficient to demonstrate SWEET’s advantages*.
>
> As shown in Re_Tab.1 (in the anonymous link), SWEET consistently achieves substantial improvements across model scales, demonstrating consistent gains even on larger models.
> >**Q2: Performance not Strong Compared to Baselines**
>
> We clarify the comparison settings: DeiT-Ti achieves 72.2% after **300 epochs** of training, whereas ViT-Ti reaches 78.22% following **pre-training on ImageNet-21K** for 300 epochs and subsequent fine-tuning on ImageNet-1K for 20k steps at **a resolution of 384×384**.
>
> In contrast, the classification results reported in Table 1 are obtained at a resolution of 224×224 with **only 30 epochs of fine-tuning** on ImageNet-1K.
> This is **sufficient to evaluate convergence speed and the effectiveness of knowledge-transfer-based initialization**, without requiring full training for 300 epochs, which would incur substantial computational cost.
>
> To further address concerns regarding the relatively lower numerical values in our experiments, we provide **extended training curves for SWEET** in Re_Fig.1.
> With longer training, SWEET achieves improved performance across various vision tasks while maintaining a clear advantage over the baseline.
>
> >**Q3: Missing Implementation Details**
>
> We appreciate your comments and will further clarify the experimental details in the revised version:
> - The ablations reported in Tables 4 and 5 correspond to classification experiments on ImageNet-1K, with Top-1 Accuracy used as the evaluation metric.
> - For detection and segmentation, since our focus is on evaluating the initialization capability of our method, **we did not perform extensive architecture-specific design** for these tasks.
> Thus, we strictly followed the original MAE settings, keeping all heads and decoders unchanged for a fair comparison. Specifically, for object detection, we employ Mask R-CNN with a Feature Pyramid Network (FPN) neck and standard RPN/ROI heads, whereas for semantic segmentation, we adopt UperNet as the decoder.
> - FLOPs are computed at a resolution of 224×224 for classification.
>
> >**Q4: Ablation on Tucker-based Constraints**
>
> We note that the ablation experiments in Table 4 **focus solely on the type of constraints**, aiming to demonstrate the flexibility of the Tucker constraint relative to the Kronecker constraint.
> **A direct comparison between SWEET and WAVE can be found in Table 1 and Table 2**.
>
> It is important to emphasize that the differences between SWEET and WAVE **go beyond the choice of constraint**.
> Beyond differences in weight-template training (SWEET’s MAE vs. WAVE’s supervised training), SWEET introduces **a width-wise stochastic scaling mechanism within the template, mitigating WAVE’s limitations in initializing models of varying widths**.
> This mechanism allows templates to adaptively adjust their dimensions to accommodate models of different sizes, thereby improving compatibility with downstream tasks across a wide range of configurations.
>
> To fairly assess SWEET’s contribution relative to WAVE, we conduct an ablation on both the type of constraints and the width-wise stochastic scaling mechanism (see Re_Tab.6).
> - Thanks to the MAE training strategy, the self-supervised weight templates learned by SWEET exhibit consistent foundational advantages across various vision tasks.
> - Notably, SWEET’s Tucker-based constraints, which support both depth and width scaling, **together with width-wise stochastic scaling, enable flexible width initialization**.
> In contrast, Kronecker-based constraints handle only depth scaling and suffer significant performance degradation with width variations.
>
> Furthermore, we provide a comprehensive comparison between WAVE and SWEET **on WAVE’s benchmark** to further highlight SWEET’s performance gains relative to WAVE (see Reviewer N7Qu(Q2)).

---

> > ### Author Rebuttal · Reviewer_Qw8q · 2026-04-01
> >
> > My concerns are addressed and I would like to raise the score to weak accept.

---

> > > ### Author Response · Authors · 2026-04-01
> > >
> > > Hi Reviewer,
> > >
> > > Thank you very much for your positive feedback and for generously increasing the score to *weak accept (4)* — we truly appreciate your recognition and support 😊
> > >
> > > We just wanted to kindly check in on one small detail: it seems that the **updated score might not have been saved in the system** (there are currently no edited comments or score changes visible on our side). We completely understand this could simply be an oversight, but we would be very grateful if you could kindly confirm or update it when convenient 🙏
> > >
> > > Thank you again for your time and support, and we hope you have a wonderful day!
> > >
> > > Best regards

---

### Official Review · Reviewer_N7Qu · 2026-03-17

**Soundness:** 2
**Presentation:** 2
**Significance:** 2
**Originality:** 2
**Overall Recommendation:** 4
**Confidence:** 4

**Summary:**

This paper proposes SWEET, a self-supervised framework for scalable vision model initialization. SWEET learns a shared weight template and lightweight scalers that can reconstruct models for different depth and width. It combines three main ideas: a unified weight matrix over heterogeneous ViT components, Tucker-based constraints, and width-wise stocastic scaling. This paper achieves consistent gains over prior scalable initialization baselines across multiple downstream tasks.

**Compliance With Llm Reviewing Policy:**

Affirmed.

**Final Justification:**

Regarding the concern about backbone models (W1), the new results on ConvNeXt-v2 are appreciated. They demonstrate that the proposed SWEET framework is not strictly limited to Transformers and can be adapted to hybrid/convolutional architectures through tensor concatenation and reorganization. Thus, I will raise the score to weak accept.

**Key Questions For Authors:**

Is the scaler still effective when the task and structure differences are greater?

**Limitations:**

There is no limitations and potential negative societal impact of their work.

**Strengths And Weaknesses:**

Strengths:
1. This paper propose SWEET, a method for initialing variable-size vision models without pre-training separate backbones.
2. This method is well motivated and coherent.
3. Also, the paper tested on multiple tasks including image classification, detection, segmentation and reconstruction, and reports improvements across mlutiple scaled ViT settings.

Weaknesses:
1. The method is overclaimed. This paper argues for size-agnostic and task-agnostic initialization, but experments is limited in a single ViT-B/16 backbone. The evidence only supports effectiveness over ViT family. More backbone models are needed.
2. There lacks ablation study that weather the benefits come from the "template mechanism" rather than stronger MAE pretrain or archetecture enhancement. Comparisons between WAVE and SWEET are needed.

---

> ### Author Rebuttal · Authors · 2026-03-31
>
> Dear Reviewer N7Qu,
>
> We sincerely appreciate your recognition of our motivation and comprehension.
> Below, we provide our detailed response.
>
> > **Q1: More Backbone Models beyond ViT**
>
> We clarify that our claims in the paper are that SWEET is task-agnostic and size-agnostic. Here, **size-agnostic refers to supporting different model scales (e.g., width and depth variants) within the same architecture** via shared weight templates, rather than being architecture-agnostic, as different architectures require separate pretraining due to their distinct parameter spaces.
> Our experiments cover initialization across **multiple ViT scales** and transfer to **diverse vision tasks** (e.g., classification, segmentation, and detection), and therefore do not constitute overclaiming.
>
> Notably, our method is not limited to ViT and **is applicable to diverse architectures**. As suggested, **we further validate it on ConvNeXt-v2**, a representative hybrid architecture combining depthwise convolutions with FFN-like 1×1 pointwise convolutions, to which SWEET can be straightforwardly extended.
>
> Briefly, a ConvNeXt-v2 consists of $S$ stages, where stage $s$ contains $L_s$ blocks with channel dimension $C_s = C \cdot 2^{s-1}$ (with $C$ as the base channel dimension). Each block comprises a $7\times7$ depthwise convolution with weights $W_d^s \in \mathbb{R}^{C_s \times 1 \times 7 \times 7}$ and two $1\times1$ pointwise convolutions with weights $W_{p1}^s, W_{p2}^s \in \mathbb{R}^{4C_s \times C_s}$.
> We aggregate all depthwise convolution weights across stages into a single unified matrix:
>
> $W_{\text{unified}} = \text{Concat}(W_d^1, W_d^2, \dots, W_d^S) \in \mathbb{R}^{(P \times C \times 49)}$, where $P = \sum_{s=1}^{S} 2^{s-1} L_s$.
>
> The $1\times1$ pointwise convolutions, which are mathematically equivalent to the linear FFN layers in ViT, are treated using the same template-based formulation, enabling seamless extension of SWEET to ConvNeXt-v2.
>
> We conduct **a single pre-training on ConvNeXt-v2-Tiny and evaluate its transfer across five size variants**, with the ImageNet-1K classification results summarized below:
>
> |Method|atto-L4|femto-L5|pico-L6|nano-L7|tiny-L9|
> |-|-|-|-|-|-|
> |WT-Select|53.24|58.78|63.61|66.62|69.42|
> |SWEET|**55.96**|**63.09**|**68.66**|**72.17**|**74.99**|
>
> These results further demonstrate that SWEET generalizes beyond ViT architectures and can be effectively adapted to diverse vision tasks and backbones through appropriate tensor reformulations.
> >**Q2: Ablation Study on Template Mechanism**
>
> We have reported ablations on constraint type in the paper (see Table 4), **comparing Kronecker- and Tucker-based templates under the same MAE pre-training and architectural enhancements**, highlighting that the intrinsic flexibility and adaptability **arise from the Tucker-based template itself**.
>
> Additionally, Table 5 in the paper presents ablations on our pre-training strategy and architectural enhancements, highlighting the effectiveness and flexibility of the template-based initialization mechanism.
>
> To further compare WAVE and SWEET under identical conditions, **we implement SWEET on WAVE’s benchmark without any enhancements to the training procedure or architecture.**
> The ImageNet-1K classification are summarized below:
>
> |Method|tiny-L4|tiny-L6|tiny-L8|tiny-L10|tiny-L12|small-L4|small-L6|small-L8|small-L10|small-L12|Pre-training Cost (Wall-clock Time)|
> |-|-|-|-|-|-|-|-|-|-|-|-|
> |WAVE|58.64|63.16|65.38|66.59|67.28|68.87|72.69|74.06|74.88|75.29|34h+64h|
> |SWEET|**58.68**|**64.26**|**66.49**|**67.90**|**68.47**|**69.08**|**72.98**|**74.63**|**75.10**|**75.56**|64h|
>
> Notably, **SWEET achieves superior accuracy across all model scales**.
> Leveraging width-wise stochastic scaling, it requires **only a single pre-training to support all depth and width variants**, whereas WAVE requires separate pre-trainings for each width scale, resulting in longer wall-clock time (34h) for constraint-based pretraining.
> These results underscore the effectiveness and efficiency of our Tucker-based template approach.
>
> >**Q3: Scaler Effectiveness with Greater Task/Structure Differences**
>
> Our SWEET effectively handles variations across both tasks and model scales, even under substantial differences.
> - For **task variation**, we evaluate on classification, detection, segmentation, and generation (Tables 1–3), demonstrating robustness across diverse vision tasks.
> - For **model scale variation**, the stochastic scaling mechanism regularizes the template during pre-training, ensuring robustness to changes in depth and width.
>
> Regarding **model architecture variation**, we supplement our experiments with ConvNeXt-v2 (see Q1), demonstrating SWEET’s generalization beyond the ViT family.
> >**Limitation**
>
> We will include an explicit "Limitation" section discussing **SWEET’s adaptability across architectures** and briefly outline future work on **extending it to T2I benchmark** evaluation for image generation.

---

> > ### Author Rebuttal · Reviewer_N7Qu · 2026-04-06
> >
> > I thank the authors for their detailed rebuttal and for providing the additional experimental data requested during the initial review.
> > Regarding the concern about backbone models (W1), the new results on ConvNeXt-v2 are appreciated. They demonstrate that the proposed SWEET framework is not strictly limited to Transformers and can be adapted to hybrid/convolutional architectures through tensor concatenation and reorganization.
> > Thus, I will raise the score to weak accept.

---

> > > ### Author Response · Authors · 2026-04-06
> > >
> > > Dear Reviewer N7Qu,
> > >
> > > We sincerely appreciate your thoughtful evaluation and are glad that our rebuttal has addressed your concerns. We are also grateful for your positive reassessment and the increased score 😊.
> > >
> > > We noticed that your acknowledgment is marked as **(b) Partially resolved**, but **no further follow-up questions were provided**. If there are any remaining concerns or aspects that require clarification, please feel free to let us know—we would be happy to provide additional details.
> > >
> > > Best regards

---

### Decision · Program_Chairs · 2026-04-30

**Decision:**

Accept (regular)

**Comment:**

All reviewers agree that the paper is well motivated and backed by strong experiments. The Tucker decomposition on the entire weight is novel and work with different kind of architecture. Regarding the reviewer consensus I recommend acceptation of the paper.